# Sub-seasonal Variability in the Boundary Layer Sources for Transport into the Tropopause Layer in the Asian Monsoon Region

Bin Chen<sup>1</sup>, Bärbel Vogel<sup>2</sup>, Xiangde Xu<sup>1</sup> and Shuai Yang<sup>3</sup>

<sup>1</sup>State Key Laboratory of Severe Weather, Chinese Academy of Meteorological Sciences, Beijing 100081, China
 <sup>2</sup>Forschungszentrum Jülich, Institute of Energy and Climate Research – Stratosphere (IEK-7), Jülich, Germany
 <sup>3</sup>Institute of Atmospheric Physics, Chinese Academy of Sciences, Beijing 100029, China

Correspondence to: Bin Chen (chenbin@camscma.cn)

**Abstract.** The Asian summer monsoon (ASM) is associated with an upper-level anticyclone and acts as a well-recognized conduit for troposphere-to-stratosphere transport. The Lagrangian dispersion and transport model FLEXPART forced by ERA-Interim data from 2001-2013 was used to perform climatological modeling of the summer season (May-July). This study examines the properties of the air

- mass transport from the atmospheric boundary layer (BL) to the tropopause layer (TL), with particular focus on the sub-seasonal variability in the tracer-independent BL sources and the potential controlling mechanisms. The results show that, climatologically, the three most impactful BL source regions are northern India, the Tibetan Plateau, and the southern slope of the Himalayas. These regions are consistent with the locations of sources identified in previous studies. However, upon closer inspection, the
- different source regions to the BL-to-TL air mass transport are not constant in location or shape and are strongly affected by sub-seasonal variability. The contributions from the Tibetan Plateau are most significant in early May but decrease slightly in mid-May to mid-June. In contrast, the contributions from India and the southern slope of the Himalayas increase dramatically, with peak values occurring in mid-July. Empirical Orthogonal Function (EOF) analysis provides further evidence that the BL sources
- in the ASM region vary across a wide range of spatiotemporal scales. The sub-seasonal behavior of these BL sources is closely related to the strength of persistent deep convection activity over the northern Bay of Bengal and its neighboring areas.

#### 1 Introduction

During the boreal summer, the off-equatorial Asian monsoon circulation in the upper troposphere and lower stratosphere (UTLS) features a unique persistent anticyclone, referred to as the Asian summer monsoon (ASM) anticyclone or South Asia high. The

5 ASM anticyclone is associated with a region in which surface emissions have been shown to enter the lower stratosphere in the Northern Hemisphere (e.g., Bannister et al., 2004; Fu et al., 2006; Randel and Park, 2006; Park et al., 2009;Randel et al., 2010; Wright et al., 2011; Vogel et al., 2014; Vogel et al., 2015; Müller et al., 2016).

Because Southeast Asia and its neighboring regions are characterized by increasing

- surface emissions and the densest population in the world, both the anthropogenic pollutants and other radiative active species such as water vapor emitted from these regions may be transported into the Northern lower stratosphere, thereby strongly affecting the radiative forcing associated with global climate change (Forster and Shine, 1997;Hegglin et al., 2010;Solomon et al., 2010;Scheeren et al., 2003;Deng et
- al., 2008;Vogel et al., 2014). Satellite data have provided evidence of a pronounced chemical signature associated with enhanced troposphere-sourced trace gas species and other pollution and burned biomass tracers in the ASM anticyclone in the UTLS (Randel and Park, 2006;Park et al., 2008;Randel et al., 2010;Bian et al., 2012). In this context, meteorologists have recently begun to pay considerable attention to the transport
- processes for atmospheric transport from the boundary layer (BL) to the tropopause layer (TL) in the vicinity of the anticyclone and the associated underlying mechanisms (e.g., Li et al., 2005;Park et al., 2008;Gettelman et al., 2011;Rauthe-Schöch et al., 2016;Müller et al., 2016;Randel et al., 2016; Pan et al., 2016).
- Although considerable efforts have been made to explore these issues, the properties of the BL-to-TL transport over the ASM region remain unclear. In particular, as a fundamental property of the transport process, the primary BL source regions and their relative degrees of importance have long been debated (Fu et al., 2006;Wright et

al., 2011;Bergman et al., 2013;Orbe et al., 2015;Yan and Bian, 2015;Garny and Randel, 2016;Pan et al., 2016;Chen et al., 2012;Heath and Fuelberg, 2014). In general, the previous studies related to BL source identification in the ASM region can be roughly divided into two categories: tracer-independent Lagrangian trajectory

- analysis and chemical tracer-based numerical analysis. As Lagrangian analysis studies, Fu et al. (2006), Wright et al. (2011), and Heath and Fuelberg (2014) emphasized the regions of the Tibetan Plateau (TP) and the southern slope of the Himalayas as the most important BL sources transporting air into the lower stratosphere. Based on a perspective of the whole ASM region, Chen et al. (2012) found that the Tibetan
- Plateau is less important than the western Pacific as a source region for tracers in the TL. Bergman et al. (2013) found that air masses originating from the Tibetan Plateau and India/SE Asia are the most important. Above mentioned efforts were made with different periods, different mean vales or with different criteria for the region of the Asian monsoon anticyclone (e.g., pressure level, potential temperature, different
- latitude-longitude regions). It is therefore difficult to directly compare the different results.

In addition to trajectory analysis, numerous studies have utilized carbon monoxide (CO) as a tracer to identify the BL sources. For example, Park et al. (2009) showed that the main surface sources of CO in the ASM anticyclone are India and Southeast

- Asia and that the contributions from the Tibetan Plateau are weak due to the lack of significant surface emissions in this region in their model simulations. Yan et al. (2015) further demonstrated that the CO within the anticyclone mostly comes from India and that little comes from East China. More recently, Pan et al. (2016) stated that the boundary layer air is primarily uplifted along the southern edge of the Tibetan Plateau and
- the region including northern India, Nepal and Southwestern China. As it stands, most previous studies are restricted to case studies or relatively short time periods and do not evaluate the spatiotemporal evolution of BL sources in sufficient detail. The ASM anticyclone is not static; instead, it oscillates in strength,

shape and position throughout the summer monsoon season (Zhang et al., 2002; Vogel et al., 2015; Nützel et al., 2016;). Recently, Vogel et al. (2015) found that the contribution of different boundary source regions to the ASH strongly depends on its sub-seasonal variability and is therefore more complex than previously believed. Pan

- et al. (2016) further highlighted that the sub-seasonal dynamics of the ASM anticyclone are an important driver of UTLS chemical transport. The results of these studies based on chemical tracers depend heavily on the distribution of the tracer sources (Park et al.,2009). For instance, due to the lack of surface emissions, the contributions from certain sources regions, such as the Tibetan Plateau and the
- southern slope of the Himalayas, are likely seriously underestimated. Therefore, a tracer-independent evaluation of the sub-seasonal variability in BL sources and the associated controlling mechanisms is crucial for improving our understanding of the distribution and variation in the chemical composition of the ASM anticyclone in the UTLS.
- Therefore, we use the day-to-day BL source locations identified from the BL-to-TL transport to examine the spatiotemporal evolution of convection sources at the sub-seasonal scale in this study. This study therefore focuses on the following questions in particular:
  - 1. From which geographical region does the air reaching the TL over the ASM
- region originate, particularly from a climatological perspective?
- 2. What are the main features of the sub-seasonal variability in these BL sources?
- 3. Are the sub-seasonal variability and the atmospheric thermal or dynamical situations closely related?

To answer these questions, we performed multi-year Lagrangian simulations using the three-dimensional transport and dispersion model FLEXPART (Stohl et al., 2005) and summertime (May-Aug) ERA-Interim data (Dee et al., 2011) from the European Centre for Medium-Range Weather Forecasts (ECMWF) for 2000-2013. By applying

a 3D labeling technique to the kinematic trajectories, a large ensemble of selected BL-to-TL trajectories (defined as those leaving the BL and reaching the TL during their transport) was built, allowing us to conduct a tracer-independent evaluation of the primary BL-departing source regions. The analysis of 6-hourly BL source

distributions is used to explore the sub-seasonal variability and associated mechanisms.

The remainder of this paper is organized as follows. Section 2 details the data, trajectory model setup, and methods used to identify the BL source regions. Section 3 presents the main results, including the climatologic state of the geographic sources,

the sub-seasonal variability, and the possible association with the atmospheric conditions, followed by a conclusions and some remarks in Section 4.

# 2 Data, model, and methodology

# 2.1 Trajectory model and its configuration

Lagrangian dispersion and transport models that track large numbers of air parcel

- trajectories are well suited for exploring the transport of trace substances, air mass sources and sinks, and the influence of transport on the atmospheric composition of the UTLS (Berthet et al., 2007;James et al., 2008;Hoor et al., 2010;Orbe et al., 2015;Vogel et al., 2016). This study uses the 3D Lagrangian particle transport and dispersion model FLEXPART. This robust model was originally designed to calculate
- the long-range and mesoscale dispersion of point source-generated air pollutants, such as those emitted by a nuclear power plant accident (Stohl and Seibert, 1998; 2005). The model calculates the trajectories of so-called tracer particles using the mean winds interpolated from analysis fields and parameterizations representing turbulence and convective transport (Forster et al., 2007). These small-scale processes, which are
- not included in standard trajectory models, are important for realistically simulating trace substance transport (Stohl et al., 2005). The inclusion of these factors makes the calculations more computationally demanding and the statistical analysis of the model

results more complex. In general, the FLEXPART model accurately simulates long-range mesoscale transport, diffusion, and the radioactive decay of the released tracers (Stohl et al., 2005).

Similar to our previous work (e.g., Chen et al., 2012a; 2017), a "domain-filling"

- 5 method is used for the initialization of the FLEXPART model. The majority of the atmospheric column above the ASM region (0°N–60°N and 0°E–160°E) is divided into 2.2 million homogeneous air parcels, each representing an equal mass (approximately 1.12×10<sup>12</sup> kg) with annual changes in the total atmospheric mass. The period of simulation is from 00:00 UTC April 15 to 00:00 UTC August 31 for each
- 10 year; however, the analyses are based on the model output for May 1 to July 31. The FLEXPART model output, including the particle ID numbers, 3D spatial positions (i.e., latitude, longitude, and height above ground), and other physical interpolated meteorological parameters (temperature, specific humidity, air density, atmospheric BL height and the tropopause height at the position of the particle) were recorded at
- 6-hour intervals (03, 09, 15 and 21 UTC).

#### 2.2 Data

The FLEXPART model can be controlled by meteorological input data generated by a variety of global and regional models. Here, ERA-Interim reanalysis data (Dee et al., 2011) from the European Centre for Medium-Range Weather Forecasts (ECMWF) are

20 used as meteorological field inputs to feed the FLEXPART model. The ERA-Interim includes a 4D variation assimilation system and produces a grid resolution of  $0.75^{\circ} \times 0.75^{\circ}$  longitude/latitude at 6-hr intervals (00, 06, 12 and 18 UTC), with 60 vertical levels from the ground to 0.1 hPa.

Additionally, to represent the strength of convection activity, daily outgoing longwave radiation (OLR) at a  $2.5^{\circ} \times 2.5^{\circ}$  horizontal resolution during 2001-2013 derived from

25 radiation (OLR) at a  $2.5^{\circ} \times 2.5^{\circ}$  horizontal resolution during 2001-2013 derived from the National Oceanic and Atmospheric Administration (NOAA) polar-orbiting satellites (Liebmann, 1996) is used as a proxy for convection.

# 2.3 BL source identification

This study pays special attention to air mass transport from the atmospheric planetary BL to the TL over ASM anticyclone regions ( $0^{\circ}N-50^{\circ}N$  and  $20^{\circ}E-160^{\circ}E$ ). A Lagrangian approach is adopted to identify BL-to-TL events. Thus, the sub-ensemble

- BL-to-TL trajectories are selected and rigorously defined as those trajectories with altitudes below the BL height at any time step that then cross the tropopause within the ASM anticyclone during their forward tracking during the period from May 1 to August 31. The BL heights are calculated by the model using the critical Richardson number concept following the approach of Vogelezang and Holtslag (1996). The
- tropopause in this study is a combination of the dynamical (2PVU) and thermal tropopause (Stohl et al., 2005). Additionally, to filter out "transient" transport events, we apply a residence time criterion to the transport trajectory selection (Wernli and Bourqui, 2002). This criterion requires a trajectory to remain within the stratosphere for at least 24 h after crossing the tropopause. Thus, only "significant" BL-to-TL
- trajectories are taken into account.

Similar to the approach adopted in previous studies (Berthet et al., 2007; Chen et al., 2012), the selected BL-to-TL trajectory ensembles are used to identify the source regions for air transport from the BL to the TL within the ASM anticyclone and neighboring regions. The BL source locations are back-calculated from the first

- encounter of a BL-to-TL trajectory with the planetary BL. Building on these large ensemble data, we can create a well-resolved density field by aggregating all the trajectories into the appropriate longitude-latitude "bin" and can then investigate the relative importance of different geographical regions as sources of BL air in the TL and their spatial and temporal distribution on a regional scale. As stated in Berthet et
- al. (2007), the BL-to-TL trajectories accurately represent the BL-to-TL air mass transport during the simulation period and act as a measure of the "density" of particles leaving the BL. Thus, this information can be used to quantitatively compare the relative importance values of different source regions.

# 2.4 Statistic analysis methods

Empirical Orthogonal Function (EOF) analysis is used here to quantitatively investigate the variability in the spatial-temporal patterns of BL sources. In this study, the daily BL source data were used to examine the spatial-temporal variability.

- Because the BL sources vary obviously over southeastern Asian and its neighboring regions (as shown in Fig. 1), we highlighted the BL source patterns prior to calculating the EOFs. First, to eliminate the effects of noise resulting from the less important regions, only regions with variance values larger than 1.0 (Fig. 1b) were considered here. Additionally, BL sources anomalies were computed by subtracting
- the May-June-July means from the total field at each grid point for each year. We examined the three leading EOFs and used their associated principal components (PCs) to identify when large-amplitude patterns in the BL sources and other variables. A number of steps are required to form composites analysis. This study selected the top 50 values of high and low PCs to assign selected events. Specifically, the
- atmospheric conditions of 50 days with high PCs are compared with those of 50 days with low PCs. After the high and low values are chosen, their corresponding atmospheric dynamical and thermal conditions, including wind fields at 850 hPa, convective available potential energy (CAPE) and OLR, are averaged. The low values are subtracted from the average of the high values to form the composite. Finally, the
- statistical significance is determined using a two-tailed Student's t-test. In addition, wavelet transform can be used to effectively extract hidden frequency-based information from the raw signal, which is normally present in the time domain. Here, Morlet wavelet analysis is used to analyze the source variability for certain target regions. The mathematical steps, algorithms and software packages
- have been thoroughly described and are readily available, making this an easy-to-use technique for climate data analysis.

#### **3 Results and discussion**

# 3.1 Climatological features of BL sources

# 3.1.1 Summer seasonal mean

To provide a climatological perspective of the BL sources, we first present the

- summer mean of BL sources over the ASM region prior to analyzing the sub-seasonal variability. The averaged density fields for all the BL-to-TL trajectories in the BL in the 1°×1° longitude-latitude bin for the summer season (MJJ) for 2000-2013 are presented in Fig. 2. As mentioned in Section 2, the density field shows the BL source distribution and the relative importance of different geographical regions as sources
- for BL-to-TL transport. Climatologically, the source regions include vast regions of southern Asia, central-east China, the western Pacific, and even eastern Africa. However, as expected, the most influential sources, i.e., those with larger values, are concentrated in regions with strong convective activity: India and its northern areas, the Tibetan Plateau, and the northern Indochinese Peninsula. This distribution pattern
- does not match those of previous studies that used CO as a chemical tracer (Park et al., 2009;Vogel et al., 2015;Yan and Bian, 2015; Pan et al., 2016).
  The seasonal BL sources presented here are largely consistent with those that have been found in previous tracer-independent studies by Wright et al. (2011) and Bergman et al. (2013), both of which identified the Tibetan Plateau and southern Asia,
- as well as their neighboring regions, as source regions. A closer look also reveals that the BL sources identified here differ from some previous studies based on trajectories analysis. For example, Fu et al. (2006) and Heath et al. (2014) stressed the importance of the Tibetan Plateau and the southern slope of the Himalayas, whereas Vogel et al. (2015) argued that the contributing sources are primarily located in Southeast Asia,
- India, and eastern and southern China. However, these studies are limited to short time periods or are case studies, which may hinder the ability to draw more general conclusions. Here the inter-annual variability is taken into account to draw more general conclusions from a climatological perspective.

The western Pacific also acts as a BL source but is less important than previously described in our earlier study (Chen et al., 2012). Obvious differences are present between our two analyses: the present analysis was restricted to BL-to-TL trajectories mostly within the ASM anticyclone, whereas the previous analysis focused on the entire

- ASM region. Additionally, the analysis here utilized the ECMWF interim analysis, whereas the previous study was based on the NCEP/GFS analysis. The differences in vertical velocity between these two data sets might give rise to the discrepancies between the two analyses, but further analysis of the mechanisms is beyond the scope of this study. Compared to the previous studies restricted to shorter time periods, the 13-year
- climatologic BL source distribution (Fig. 2) provides a better quantitative description of the geographic source regions associated with the transport of BL air mass to the UTLS in the ASM region, which presumably influences the chemical nature of the atmosphere in this region.

# 3.1.2 Variance and coefficient of variation

- The coefficient of variation (CV) and variance for the BL-to-TL sources are calculated from the daily time series for each grid cell and are shown in Fig. 1. Because the CV is a measure of the dispersion of data points in a data series around the mean, small values are associated with source regions with little variability in their contributions, whereas large values are associated with regions that are only
- infrequently sources. The most impactful BL-to-TL source regions have CV values of less than 3.0 (Fig. 2) and are also characterized by large variance values (Fig. 1b), indicating marked variability. Based on Figs. 2a and 2b, the most influential BL source regions of large uplift identified in this study are the Indian peninsula, northern India, the Tibetan Plateau and the northern Bay of Bengal. These areas not only
- contribute significantly to the BL-to-TL transport during the summer season but also feature distinctive sub-seasonal variability.

# 3.1.3 Time series for air mass transport

To provide a quantitative overview, the daily evolution of BL-to-TL uplift for each year from 2000-2013 is calculated by counting the number of tropopa