# Peer review of "Sub-seasonal Variability in the Boundary Layer Sources for Transport into the Tropopause Layer in the Asian Monsoon Region"

_Atmospheric Chemistry and Physics, 2017_

## Referee Comment (RC1) · Anonymous Referee #1 · 13 Apr 2017

General comments:

This paper is not publishable in its present form. The first half of the analysis (Secs. 3.1.1 to 3.2.2, pages 9-14) represent a positive contribution. This analysis focuses on climatological aspects of air transport from the boundary layer into the Asian Summer Monsoon anticyclone, providing valuable insights into robust characteristics of this transport. The analysis in the second half (Secs. 3.2.3 to 3.3.2, pages 14-18), however, is too weak to publish. In particular,

- Regarding Sec. 3.3.2: The conclusions drawn from this analysis are not consistent with the figures shown. I see no 'distinct' peak in variability at 10-20 day periods (as claimed by the authors). There is a hint of a peak near 32 days for India, but otherwise

there are no distinct peaks at 30-60 day periods either. The largest peaks correspond to 64-128 day periods – considering that you use 3 months of data (each year), those peaks might indicate nothing more than the seasonal cycle. In creating the multi-year time series, did you combine a series of summers, for example, placing May 1 2001 immediately after July 31 2000? That practice is problematic and should be avoided.

- Regarding Sec. 3.3.1: For EOF analysis to be effective and the associated spatial patterns to be meaningful, the case for dominant modes should be stronger. Explained variances of 10-15% for each of the first three modes is small and the separation between these modes (and higher modes as well) might not be statistically significant (See North et al 1982, Monthly Weather Review, 'Sampling errors in the estimation of Empirical Orthogonal Functions'; there is also a review in Journal of Climate, Monahan et al 2009; 'Empirical Orthogonal Functions: The Medium is the Message'). That this section does little more than describe the spatial patterns and temporal variability of the first three EOFs makes it particularly important that these EOFs be dynamically meaningful and independent.

- Regarding Sec. 3.3.2: This analysis does little more than confirm that EOFs 1, 2, 3 represent similar modes of variability and that none stand alone as being dynamically relevant. This section is mostly descriptive and does not actually contribute to conclusions drawn at the end of the section [e.g., that the anti-cyclone is a bubble].

Specific Comments:

Page 4, Line 8: '. . . due to the lack of surface emissions, the contributions from certain sources regions, such as the Tibetan Plateau and the southern slope of the Himalayas, are likely seriously underestimated'. Do you mean 'lack of surface emission data'? If there was a lack of emissions (as stated) how could sources be underestimated (i.e., no value underestimates zero)?

P 6 L 4-15: Please specify the initial parcel spacing (i.e., how many degrees/kilometers between parcels) and how often parcels are initiated. In particular, be sure to discuss

how the model set up affects the results. For example, does the model setup maintain a uniform distribution of BL parcels throughout the entire analysis period?

P 6 L 24-27: Please discuss how the distribution of convection parameterized by FLEX-PART compares with OLR.

P 7 (Sec 2.3): The question above (P 6 L 4-15) regarding the distribution of BL sources could be addresses in this section. It is not clear from your discussion that parcel distribution in the boundary layer is uniform. If that distribution is not uniform (throughout the analysis period), then the BL-TL statistics are not reliable measures of transport.

P 8 L 2-12: Provide a reference for the EOF analysis used and a brief description of the information that it allows you to extract.

P 8 L 7-9: Please describe (more precisely) how the low variance regions were filtered from the data. Why not use space-time averaging to reduce noise variance? How large of an impact does noise have if it is not filtered out?

P 8 L 9-10: What is meant by 'total field'? E.g., were monthly anomalies calculated by subtracting monthly means from the overall time mean?

P 8 L 21-26: Provide references for the wavelet methods used. Secs. 3.1.1 and 3.1.2: These two sections along with Figs. 1 and 2 contain a lot of redundant information. These two sections should be condensed into a short discussion of a single figure – Fig. 2 would be a good choice.

P 11 L 8-9: The text mentions an increase of mass transport from May 1 to July 31 – Fig. 3 shows mass transport from June 1 to August 31 (i.e. May 1 to May 31 is not shown). Please correct this discrepancy. A related question: is the time of transport related to the time of tropopause crossing? (Do the times along the x-axis of Fig. 3 represent the time of tropopause crossing?) Or some other reference.

P 12 L 22-23: You state that the air mass uplift from India (in Fig. 6a) increases dramatically before June 15 then decreases slowly afterward. To me, it looks like the

descent is just as dramatic as the ascent (absolute values of the slopes before and after June 15 are similar).

P 12 L 25-27: The meaning of the sentence 'Therefore, compared . . .' is too obscure.

P 13 Paragraph 1: This paragraph (regarding contribution by the Tibetan Plateau to BL-TL transport) is too unclear, unnecessarily speculative, and contains apparent discrepancies. For example:

- You state that 'the large contribution from the TP source in early May could be due to the strong sensible heating' and that the decrease in June 'presumably coincides with the decrease in sensible heating in this region'. The seasonal timing of sensible heating (e.g., from reanalysis data) should be cited here.

- You also refer to the seasonal timing of convection. You should refer explicitly to the OLR data set you use.

- You claim that the 'rain belt' reaches the Tibetan Plateau in mid-July and suppresses convection. My understanding is that Summertime rain in mountainous regions is associated with convection. Please clarify why the rain belt suppresses convection.

P 13 L 20 – P 14 L 2: Could the West Pacific contribution be related to tropical cyclone activity – again, the OLR data set can be helpful here.

P 14, last paragraph of Section 3.2.2: It would be helpful to clarify (state explicitly) that your study examines total mass transport whereas chemical tracer studies examine the mass transport weighted by chemical concentrations. This paragraph only alludes to that fact.

Technical details:

P 2, L 7 (and other lists of references): Remove the comma before the reference year; e.g., change 'Bannister et al., 2004' to 'Bannister et al. 2004'. Add a space between ';' and next reference; e.g., change 'Park et al., 2009;Randel' to 'Park et al 2009; Randel'.

P 4, L 3: Undefined acronym (ASH) used. P 4, L 10, 15, 17: The word 'therefore' is over used. P 8, L 12: Delete 'when'. P 10 L 22: Should 'Figs 2a and 2b' be 'Figs 1a and 1b'? P 11 L 22 (also L 27, P 12 L 1): No need to place 'hotspot' in quotes. If you want to define what a hotspot is – then place the first occurrence in quotes (and only the first occurrence) followed by the definition. Fig. 8 Caption: Undefined acronym (ASH) used. P 16 L 5: Change 'EOFs' to 'EOF3' P 18 L 10: Change 'This region is overlaps . . .' to 'This region overlaps'

---

## Referee Comment (RC2) · Anonymous Referee #2 · 27 Apr 2017

Review of "sub-seasonal variability..." by Chen et al., 2017

General comment:

This paper deals with the origin of BL air masses that cross the tropopause during the Asian Summer Monsoon with a focus on its sub-seasonal variabilty. It is based on the analysis of 13 years of Lagrangian modelling. The subject of this paper is relevant to ACP and the results could bring some interesting and new information. Nevertheless, the paper is not yet fully convincing and the authors should adress some major issues before publication. These issues detailed below concern both the methodlgy and the results. Concerning the methodology, the choices made for the Lagrangian modeling should be discussed: use of kinematic versus diabatic trajectories on top of the con-

vective outflow; uncertainties related to the use of ERA-Interim reanalysis. Relative to this particular point, the large discrepancies between the present study and Chen et al. (2012) should be adressed. Concerning the results, the authors use many statistical tools such as wavelet or EOF decomposition but the methodology is not detailed enough, their added value is not straightforward and their analysis is confusing. Therefore, the presentation and the analyses of the statistical results should be improved or some parts disgarded as not conclusive.

Detailed comments:

Methodology:

1/ Lagrangian modeling: The 3D Lagrangien particle transport and dispersion computation presents some limitations that are not discussed: - The use of ECMWF ERA-Interim reanalysis may be responsible for systematic biases. In many other studies different sets of reanalyses are used and the results are compared to determine the uncertainties and strenghten the conclusions. For instance Bergman et al. (2013) compare results of trajectory calculations from ECMWF, NCEP/GFS and MERRA. They show that the analysis used do not change their conclusions. The authors of the present study have already used NCEP/GFS reanalyses to deal with the same problematic for the 2001-2009 period (Chen et al., 2012). It should therefore be possible to discuss the differences based on this previous study for the common period (2001-2009). - For vertical transport above the altitude of convective outflow, the trajectories are computed with vertical velocities from the reanalyses. A number of studies dealing with the same subject are based on vertical transport computed from radiative heating rates (Tissier and Legras, 2015, Garny and Randel, 2016). For instance, Garny and Randel show that kinematic calculations are responsible for larger vertical dispersion of the trajectories and that "diabatic calculations result in more trajectories traveling to higher levels that lie well above the tropopause". It could be interesting to discuss the impact of your choice to use kinematic trajectories on your results and conclusions. - The OLR are used to "represent the strenght of convection..." It is not clear wether

they are used to trigger convection in the model or simply to characterize convection as an external source of information. In the second case it is not obvious that the model triggers convection in good coincidence with convective activity derived from the OLR. Are there references for such a coincidence concerning FLEXPART? Otherwise is it posssible to give such evidence?

2/ ASM anticyclone boundaries: - The study is dedicated to "transport from the BL to the TL over ASM anticyclone region" and "BL-to-TL trajectories are selected and rigorously defined as those trajectories . . .. that then cross the tropopause within the ASM anticyclone". Nevertheless, the boundaries of the anticyclone are not defined rigorously nor used quantitatively. The computations are made with the anticyclone region defined as the 0-50°N and 20-160°E box instead. The longitudianl boundaries are more or less correct but the latitudinal boundaries are much too large. The AMA is centered around 25°N and extends roughly south to 15°N and north to 40°N. A 0°N south boundary probably leads to an overestimation of the imapct of southern regions such as the southern part of IN, BB, SC and more importantly from the WP box. Furthermore, the AMA is obviously not a rectangle box but an ellipsoid with boundaries undergoing a strong sub-seasonal variability as recognised in the manuscript p3l28-p4l1-6 (see also Popovic and Plumb et al., 2002, Ploeger et al., 2015). Therefore, the subseasonal variability of the origin of air masses crossing the tropopause within the AMA may vary also with the AMA variability itself. Recent studies have tried to determine the dynamical boundaries of the AMA. For instance Ploeger et al. (2015) have developped a sophisticated PV-Based criterion to follow the AMA dynamic and Barret et al. (2016) a simplified criterion based on geopotential height anomalies. Some test on limited periods (one season for instance) could be made with dynamical AMA boundaries to evaluate the impact of the AMA dynamics on the results. If the authors choose a rectangular box, they should not state "within the ASM anticyclone" but "within the region that encompasses the ASM anticyclone". In which case the southern boundary of the domain has to be revised to at least 15°N.

3/ Pollution transport: - The authors state many times that a tracer-independent Lagrangian modeling is needed to characterize the origin of the air masses affecting the UTLS composition and that pollution tracer studies are weighting the results towards polluted regions. The UTLS composition is affected by the transformation of species (gases and aerosols) emitted by anthropogenic, fire or natural sources which are region dependent. Therefore, tracer-independent Lagrangian modeling is not the best method to characterize regions that affect the most the UTLS composition. The afore mentioned statement should be modified accordingly. Another example is p14l13-14 : "the low level... led to inappropriate conclusions". This statement is exagerated and also needs to be modified. Conclusions of CO-based studies are appropriate to determine where CO and associated pollutants come from and therefore which regions are significantly contributing to modifications of the composition such as enhancements of CO, O3 or aerosols in the AMA. TB is not one of them due to the lower level of pollutants compared to N India and the southern Himalayan slopes. Of course, uplifted air masses from the TB or WP or BB modify the UTLS composition because they are wetter or O3-depleted. Inappropriate should be replaced by complementary.

4/ Statistics analysis methods: The reader needs more detailed explanations of the statistical methodology. The "BL sources anomalies . . . substracting May-June-July means from the total field...". Does this means that the May-June-July are substracted from the April 15 to August 31 ? Such a difference is not an "anomaly" but a kind of seasonal mean. Anomalies are normally defined as oultliers from means. Furthermore, the mean is not really seasonal and the choice of May-June-July is not easy to understand because the monsoon generally starts in June and ends in September. The authors use EOFs and their PCs but more details, references should be given about the calculation and signification of EOFs and Pcs. What is the meaning of high and low Pcs? What is the difference between a high and a low PC relative to the BL to TL transport?

Results:

1/ Comparison with Chen et al. (2012): - p10l1-2: It is stated that "the WP . . . is less important than in our earlier study (Chen et al. (2012)". It is the less that can be said! Fig. 2 from the present paper and Fig. 3 from Chen et al. 2012 show completly different structures. The maxima are located over northern equatorial Indian Ocean (the most important maximum), BB, South China Sea and WP for Chen et al. (2012). Here they are located over India and over the TP. Equatorial Indian Ocean has disappeared and the density over WP is 2 to 3 times less important than over IN and TP! These discrepancies should be described and discussed more thoroughfully in the present study. Indeed, the hypotheses given are rather light to explain such discrepancies: - the studied domain is smaller here (ASM region) than in Chen et al. (2012): here 0-50°N and 20-160°E and in Chen et al. (2012) 0-60°N and 0-160°E. The region north of 50°N has probably little impact on TST or BL-to-TL transport. The part west of 20°E encompasses Africa and could have some importance in the maxima over the equatorial Indian Ocean. Nevertheless, during the ASM transport is rather from Asia to Africa via the Tropical Easterly Jet, and no maxima is present over India and the TP in Chen et al. (2012). - Chen et al. (2012) have used NCEP/GFS analyses while the present study is based on ECMWF ERA-Interim reanalyses. As suggested above, a comparison should be made between (i) the wind fields from both datasets (ii) the trajectory distributions even though it should not change much from the difference between Chen et al. (2012) and the present study. Nevertheless, according to Bergman et al. (2013) the use of different analyses do not change radically the results of Lagrangian simulations. For instance, using ECMWF/NCEP and MERRA, the contribution of fresh air masses within the AMA varies from 35 to 38

2/ Use of statitiscal tools: - p10l15-26 and Fig1a/b: The CV and variance maps do not bring much information. The first one, weighted by the inverse of the density from Fig. 2 is close to its negative picture. The second one closely follows the density map of Fig 2. The text relative to these plots could be shortened and the plots could be removed. - From the analysis of the wavelet spectra (Fig. 7), the authors determine marked 10-20 and 30-60 days peak for the variability of the anomaly for the TP. The

90 days peak that probably correponds to the seasonal variability is clear but instead of marked peak, Fig. 7 rather shows a broad continuum from 10 to 60 days. For the IN region, the 30 days peak is much clearer. - The analysis of the spatial patterns of the sub-seasonal variability with the EOF is rather confusing. The analysis of the EOF focuses on the three leading EOF that explain less than 40- In 3.3.2, the authors aim at characterizing "the relationship between sub-seasonal variability and atmospheric composition" through composite analysis (p16-18). The aim is interesting but I find this part of the paper particularly confusing and I think it should be largely improved (starting by a better explanation of the method as required above). - The authors "hypothesized" that the difference between conditions corresponding to high and low PCs explain the controling mechanism of transport. The basis of this relationship is unclear to me and should be developped further. - PC1 and PC3 high – low wind composites are characterized by North easterlies and PC 2 by Westerlies. What is the meaning of these patterns in terms of monsoon weather variability and what is the link with convection ? In what sense are the composite explaining the BL-to-TL transport sub-seasonal variability are they for instance linked to enhanced or decreased transport? Why? - high CAPE and low OLR are indicative of deep convection activity. Nevertheless, the composite maps of Fig. 9, 10 or 11 do not display a clear coincidence of both. The CAPE pattern presents a very localised band of high values over the BB along the Indian east coast and the lowest OLR are over a large domain of the BB. Could the authors discuss this point?

3/ Interannual variability: The interannual variability displayed in Fig. 3a is rather strong with some years showing some values largely on top of the others. The authors comment about this plot is "This annual variability is presumably associated with... strength of the ASM". The absence of discussion is rather frustrating for the reader and Fig. 3a could be removed or discussed further even if the paper does not deal with interannual variability.

4/ Comparison with Bergman et al. (2013): Concerning the regional contributions,

comparisons with Bergman et al. (2013) should be made deeper. The IN contributions found here during the monsoon (20-40

Details: - p8l12: sentence is not clear. - p9: "this distribution pattern does not match those of previous studies that used CO as a chemical tracer". Concerning studies using CTM's (such as Park et al. 2009, Yan and Bian, 2015) they are not displaying the same distributions and quantitative comparisons are not possible. Nevertheless, there is a qualitative agreement because these studies clearly show the predominance of South Asian relative to East Asian emissions to fill the AMA with CO. Particularly the highly polluted Indo-Gangetic Plain is highlighted here as a region largely participating to BL-to-TL transport. The most important qualitative difference between CTMs studies and the present one is of course TB with high convective activity and low pollution emissions. - p10l22: Figs 1a and 1b - p11l25: the northward shift of sources and their retreat from pre to post-monsoon is also clearly demonstrated in Barret et al. (2016) from Eulerian chemistry transport simulations and satellite observations of CO distributions once again showing that both approaches are qualitatively agreeing. - p13l23: "as the TP region" - p13l27: "possibly due to . . . variability in SST". This can easily be verified with SST from ECMWF reanalyses for instance. - p15l13-14: this sentence is not really informative and could be removed.

Additional Refs: Barret et al. (2016), Upper-tropospheric CO and O3 budget during the Asian summer monsoon, Atmos. Chem. Phys., 16, 9129-9147, doi:10.5194/acp-16-9129-2016. Ploeger et al. (2015), A potential vorticity-based determination of the transport barrier in the Asian summer monsoon anticyclone, ACP, 15, 13145-13159, 2015, doi:10.5194/acp-15-13145-2015. Popovic and Plumb (2001), Eddy shedding from the upper-tropospheric Asian monsoon anticyclone, J. Atmos. Sci., 58, 93-104, doi: 10.1175/1520-0469(2001)058. Tissier and Legras (2016), Convective sources of trajectories traversing the tropicaltropopauselayer, Atmos. Chem. Phys., 16, 3383-3398, doi:10.5194/acp-16-3383-2016.

---

## Referee Comment (RC3) · Anonymous Referee #3 · 27 Apr 2017

This study presents detailed analyses on the transport source regions from the boundary layer in to the tropopause layer in the Asian summer monsoon region. A Lagrangian trajectory model simulations of more than 10 years are used to identify regions with most frequent boundary layer to tropopause layer transport. Sub-seasonal variabilities of statistics of vertical transport were also explored in detail. The results presented in this study are considered to be of interest among extended scientific community. I have a number of comments and suggestions for the authors might take into consideration.

General Comments

- The goal of this study needs to be mentioned explicitly. I am not sure if the trajectory simulations were aimed at looking at the vertical transport from the boundary layer in

to the upper troposphere in general or in to the Anticyclone. What do the trajectory simulations and their statistics represent in this study? The results seem to be based on the frequency of vertical transport from the boundary later in to the tropopause layer over a broad region instead of Asian summer monsoon anticyclone itself. I think the authors need to explain what physical processes they are going to focus on and why the trajectory model is a useful tool to use answer their questions.

- Regarding making connections between the physical processes, the authors claim that persistent deep convection over Bay of Bengal is responsible for the vertical transport. And yet, there is no evidence supporting this argument. Without showing a map of convection and its temporal variability over the monsoon region, this statement has no significance.

- Regarding the FLEXPART model simulations used in this study, there are fundamental questions to be answered before the results can be discussed. Those include, 1) What are the fundamental differences between 'tracer-independent' and 'tracer-based' analyses? What are the caveats of 'tracer-independent' method? 2) What is '3D labeling technique'? 3) What is the difference between 'kinematic' versus 'adiabatic' trajectory analysis? 4) How are the BL sources are defined and initialized? 5) How does the model handle deep convection? Is it parameterized? Those should be explained sufficiently to support the results of trajectory model simulations presented here.

- In my opinion, it should be mentioned explicitly that 'tracer-independent' method has its own limitations. If there are no surface sources where the vertical transport occurs, it would make no contribution to the upper tropospheric chemical environment.

- It is not clear the three source regions, e.g., Northern India, Tibetan Plateau, and the southern slopes of Himalayas, are emphasized in this study. I would recommend putting more focus on those regions instead of describing details over multiple geographical regions.

- Three questions listed in introduction have not been answered. The authors might

want to revisit those questions in conclusion.

- It would be helpful to include figures of actual trajectories especially over the regions where most frequent transport is occurring.

- It is hard to draw conclusions based on the EOF analyses alone without showing physical patterns of the variabilities. What are the physical processes underneath those statistical analyses?

Specific Comments

- Why the summer season is defined as May-July instead of May-Aug or Jun-Aug?

- It is unclear if the analyses period is either 2001-2013 or 2000-2013.

- P2 (L10) – population in the world [reference needed].

- P2 (L19) – meteorologists -> scientists

- P3 (L1-2) – Citations must be organized chronologically.

- P3 (L12-16) – This is a vague statement. It would be helpful to give more specific information or avoid listing all the things one can possibly think of.

- P3 (L21) – In Park et al. (2009), lack of surface emissions and also shallower convection over the Tibetan Plateau may have mentioned.

- P6 (L2) – Long-range mesoscale transport does not sound right. Long-range transport should he a large scale and mesoscale transport does not represent long-range.

- Why is the model domain restricted to 0-60N latitudes and 0-160E longitudes? And the BL source regions are chosen between 0-50N and 20-160E, which seems rather subjective. The model run is performed from April to August but only the results between May and July have been shown most of the cases. Why is that? It is not clear how the convection is parameterized or represented in the model.

- P7 (L9-10) – What is the reason for using both the dynamical and thermal tropopause

definitions here?

- P8 (L10-12) – A verb is missing in this sentence.

- P8 (L14) – I wonder how many total cases are there and why 50 high and low cases were chosen. It this statistically significant?

- Fig. 1 shows both the coefficient of variance (CV) and variance. In my understanding, CV is variance divided by the mean. So, the regions with small CV represent larger mean values. I am not sure how to interpret this result.

- Fig. 1 – Thickness of the black dashed line can be increased for better representation of the region (also in Fig. 2 as well).

- P10 (L5) – ECMWF interim -> ERA interim?

- P10 (L19-20) – that are only infrequently sources -> needs to rewrite this.

- P10 (L22) – Figs. 2a and 2b -> Figs. 1a and 1b

- P11 (L6-7) – This annual variability. . .of the ASM -> I think there needs to be a reference to this sentence. Does annual variability refer to interannual variability or seasonal variability?

- Fig. 3 (bottom) – Is the time step every 4 days? Maybe longer time steps (fewer data points) can be used to simplify this figure. Also, what do the notable peaks in each year's time series (top panel) represent? Do they represent more frequent transport? What are they correlated with?

- P12 (L2-7) – What do the authors think the reason is for those seasonal differences in the air mass transport?

- P12 (L16) – What do 'net masses' mean?

- P12 (L20) – maximum relative contribution -> maximum contribution

- P13 (L11-12) – This decrease . . . this region -> A figure or reference is needed to

support this.

- P13 (L12) – He et al. (2007) should be cited at the end of this sentence.

- P13 (L15-19) – Without showing figures of convection or references, this statement has no basis.

- Fig. 7a – There are only a few years showing 90% power. This does not represent climatological features.

- Fig. 8 – I am not convinced with the results shown here. First, the total variance explained by the three leading EOFs are only 37.1 %. Second, the PC1 seems to represent annual cycle with some added interannual variability. I am not sure how to interpret this result.

- P15 (L24) – I see a dominant negative variability instead of obviously positive values in Bay of Bengal in Fig. 8.

- P16 (L5) – EOFs -> EOF3

- P16 (L12-17) – This paragraph is somewhat confusing to me. I am not sure what the authors try to say. Should we believe the EOF results here or not?

- P17 (L2-5) – This is a vague statement. What is the hypothesis based on? Why do authors think high versus low PCs explain controlling mechanisms to some extent?

- P17 (L10-11) – Northeastern Korea (Fig. 9b) -> I am not sure where the signal is located on the map?

- P17 (L13) – The deep convection (represented by OLR anomalies) are

- P17 (L15-16) – These areas of. . .summer monsoon. -> This has not shown anywhere in the paper.

- P18 (L4) – large deep convection differences -> I am not sure what this sentence means.

- P18 (L10) – This region is overlaps -> This region overlaps

- Section 4 (conclusion) – This section needs to be rewritten for clarity. It might be better to focus on fewer geographic regions as well.

- P19 (L22) – We hope that the results of this study will provide some clarification of the. . .-> We believe the results in this study has provided clarification of the. . .